# Serum microRNA miR-491-5p/miR-206 Is Correlated with Poor Outcomes/Spontaneous Hemorrhagic Transformation after Ischemic Stroke: A Case Control Study

**DOI:** 10.3390/brainsci12080999

**Published:** 2022-07-28

**Authors:** Xindi Song, Junfeng Liu, Yanan Wang, Lukai Zheng, Ming Liu

**Affiliations:** 1Center of Cerebrovascular Diseases, Department of Neurology, West China Hospital of Sichuan University, Chengdu 610041, China; xindi_song@foxmail.com (X.S.); junfengliu225@outlook.com (J.L.); yanan_wang1105@163.com (Y.W.); 2Institute for Stroke and Dementia Research, Ludwig Maximilian University Hospital of Munich (KUM), 81377 Munich, Germany; lzheng@med.lmu.de

**Keywords:** microRNAs, ischemic stroke, biomarkers, spontaneous hemorrhage transformation

## Abstract

Background: It is unclear whether miR-491-5p, miR-206, miR-21-5p or miR-3123 are associated with functional outcomes and hemorrhagic transformation (HT) after acute ischemic stroke (AIS). In this study, we aimed to investigate the correlation between these four microRNAs and functional outcomes, as well as spontaneous HT after AIS; Methods: We included 215 AIS patients and retrospectively assayed for miR-21-5p, miR-206, miR-3123 and miR-491-5p levels in serum. Poor functional outcome was defined as a modified Rankin Scale score ≥ 3. Spontaneous HT referred to hemorrhage detected in follow-up brain imaging but not on admission, without reperfusion therapies. Logistic regression, generalized additive model and 2-piecewise regression model were used to explore the independent, non-linear correlation between miRNA expression levels and outcomes; Results: We included 215 AIS patients. Higher miR-491-5p level independently reduced the risk of poor functional outcomes at 1 year (OR 0.90, 95% CI 0.82–0.98, corrected *p* value = 0.044). Higher miR-206 level significantly increased the risk of spontaneous HT (OR 1.64, 95% CI 1.17–2.30, corrected *p* value = 0.016). There was a nonlinear correlation found between miR-491-5p level and 1 year outcome with an inflection point of 2.180, while an approximately linear correlation was observed with an inflection point of 2.037 between miR-206 level and spontaneous HT; Conclusions: Higher serum miR-491-5p level independently reduced risk of 1-year poor functional outcome of AIS patients. Higher serum miR-206 level independently increased the risk of spontaneous HT in AIS patients. These two miRNAs may be as the potential biomarkers for improving prognosis after AIS.

## 1. Introduction

Stroke is the second leading cause of death in the world and the leading cause of death in China, which accounts for a fifth of the worldwide population [1]. Approximately 70% of newly onset strokes are acute ischemic stroke (AIS), and more than half of the survivors suffer from disability [2]. Exploring the key factors is essential for clinicians to design treatments accordingly and improve the outcomes of stroke patients.

With the development of technology, noncoding RNAs, such as miRNAs and piRNAs, have attracted attention due to their prospect of becoming potential biomarkers for disease prediction and targets of intervention. A systematic review summarized the current evidence on the association between miRNAs and the diagnosis of stroke, and it is still controversial whether miRNA expression is independently associated with stroke outcomes [3].

Previous reports suggested that certain miRNAs regulate matrix metalloproteinase-9 (MMP-9) expression during stroke [4]. MMP-9 is an enzyme that digests type IV collagen, laminin, and fibronectin, which are major components of the basal lamina in cerebral blood vessels [5]. Preclinical and clinical studies indicated that increased MMP-9 expression plays an important role in blood–brain barrier (BBB) disruption, thus leading to poor outcomes after ischemic stroke [6,7]. However, the underlying mechanism of the deleterious effect of MMP-9 during stroke still lacks adequate evidence. MMP-9-related miRNAs may play a role in the regulatory mechanism of MMP-9. We have reported the effect of serum MMP-9-related miRNAs miR-21-5p, miR-206 and miR-3123 in cardioembolic stroke [8]. Here, we analyzed the cohort in our stroke registry to determine the correlation between miR-21-5p, miR-206, miR-3123 and miR-491-5p and functional outcomes after ischemic stroke. In addition, we investigated the correlation between the four miRNAs above and spontaneous HT in general AIS patients.

## 2. Materials and Methods

### 2.1. Patient Selection

AIS patients were consecutively admitted to the Department of Neurology, West China Hospital, Sichuan University, from November 2016 to September 2018. All patients received a clinical diagnosis of AIS according to World Health Organization criteria [9], and this diagnosis was confirmed by computed tomography (CT) or magnetic resonance imaging (MRI). Patients were excluded if they (1) did not have blood samples at admission, (2) had intracerebral hemorrhage (ICH) at admission, or (3) received reperfusion therapies after stroke onset. Informed consent was obtained from each patient or their relatives. This study was approved by the Scientific Research Department of West China Hospital.

### 2.2. Baseline Data Collection

We recorded baseline clinical and demographic information on admission, including age; sex; onset time; stroke severity assessed by the National Institutes of Health Stroke Scale (NIHSS) and Glasgow Coma Scale (GCS); axillary temperature; preexisting risk factors (history of hypertension, diabetes mellitus (DM), hyperlipidemia, atrial fibrillation (AF), acute myocardial infarction, valvular heart disease, ischemic stroke or transient ischemic attack, hemorrhagic stroke). Therapies used before admission, which may influence outcomes, were collected, including anti-thrombotic medications (antiplatelet agents or anticoagulants) and lipid-lowering medications. All patients in our study underwent CT within 24 h on arrival at the emergency department. Patients received another CT or MRI again within 1 week after stroke onset. Any changes in symptoms or signs were recorded. For patients with neurological deterioration, an emergency brain CT was performed. The time of each neuroimaging was collected. CT and MRI were evaluated by a neuroradiologist blinded to patient details.

### 2.3. Blood Sampling

All blood samples were collected within 24 h of admission. Venous blood was collected in coagulant-containing tubes and was then sent to the laboratory within 30 min of collection. After centrifugation (3000 rpm for 15 min at 4 °C), serum was collected and immediately frozen at −80 °C until further use.

### 2.4. Selection and Assessment of Candidate MMP-9-Related MiRNAs

TargetScan version 6.2 (www.targetscan.org, accessed on 11 December 2013), miRanda August 2010 release (www.microrna.org/microrna/home.do, accessed on 11 December 2013) and miRDB version 4.0 (www.mirdb.org, accessed on 11 December 2013) were used to predict miRNAs acting upstream of the human MMP9 gene in silico. We also reviewed previous literature to identify candidate miRNA regulators of MMP-9 expression.

### 2.5. Extraction of miRNAs, Reverse Transcription and qRT–PCR

Serum miRNAs were retrospectively isolated from serum samples using the miRNA Isolation Kit (Tiangen, Beijing, China). Before this isolation step, all serum samples were spiked with microRNA miR-39 (2.5 μL, 20 μM), which served as a reference RNA. Serum miRNA was eluted in 25 µL, among which 15 µL was used for quantitative real-time polymerase chain reaction (qRT–PCR).

We used the All-in-One™ miRNA First-Strand cDNA Synthesis Kit (20 rounds of reverse transcription) (Funeng, Guangzhou, China) for reverse transcription. PolyA was tailed at the end of miRNA 3′, after which a unique Oligo-dT adaptor primer generated cDNA of the first strain corresponding to miRNAs. Then, specific PCR primers were used to detect target miRNAs. Allin-One™ qPCR Mix including SYBR^®^ Green was used to detect the cDNA generated through reverse transcription.

The resulting solubility curve and cycle threshold (Ct) were evaluated via CFX Manager software version 2.0 (Bio–Rad, Hercules, CA, USA), and the relative expression (fold difference) of candidate miRNAs with respect to the average level in patients who did not suffer HT (NHT) was calculated using the 2^−^^ΔΔCt^ method according to the formula:fold difference = 2 ^{− [(Ct Candidate − Ct miR39-3p) − (Mean Ct CandidateNHT − Mean Ct miR39-3pNHT)]}(1)

Expression levels of miRNAs are expressed as fold difference per µL of serum, as previously described [10].

### 2.6. Outcome Measures

Our primary outcomes were poor functional outcomes at 3 months and 1 year (defined as a modified Rankin Scale score ≥ 3). Secondary outcome was spontaneous HT. Spontaneous HT was identified when hemorrhage was not detected by the initial CT but was later confirmed (within 7 days of admission) by a second CT or MRI, without using reperfusion therapies. HT was classified by the European Cooperative Acute Stroke Study (ECASS) criteria into 2 types, hemorrhagic infarct (HI) and parenchymal hematoma (PH), and then further stratified into 4 subtypes: HI1, HI2, PH1, and PH2 [11].

### 2.7. Statistical Analyses

Data were reported as the mean ± standard deviation (SD) or median (interquartile range, IQR) for continuous variables or frequencies with percentages for categorical variables. The levels of miRNA expression were natural log (Ln) transformed since the original distribution of raw miRNA expression data was skewed. A one-unit increase in Ln-miRNA refers to a natural-logarithm-fold (about 2.7-fold) increase in miRNA expression level. Data were examined for normality using the Shapiro–Wilk test, where the data did not fit the normal distribution. Univariate analysis was performed to initially explore the difference between miRNA expression levels and each outcome measure, as well as to screen out the potential confounding factors that may influence outcomes. Categorical variables were compared between groups with the chi-squared test or Fisher’s exact test when appropriate, and continuous variables were compared with the Welch t-test or the Mann–Whitney U test. Variables that showed a univariate correlation with outcomes (*p* < 0.1) or with clinical constraints were further analyzed in logistic regression models as confounding factors. Multivariable logistic regression analysis was performed to assess the independent correlation between each miRNA level and outcomes. The odds ratio (OR) and 95% confidence interval (CI) were calculated. The type I error rate was controlled using the Benjamini-Hochberg method for multiple comparisons [12].

A generalized additive model (GAM) was used to visually assess whether there were nonlinear correlations between serum miRNA expression levels and risk of outcomes. The miRNA expression levels were incorporated into the model in form of Ln-transformed values of fold difference per µL of serum. The risk of outcomes was calculated and incorporated into the model as Lg-transformed value of odds ratio, namely Lg (odds ratio). Once a nonlinear correlation was discovered, a 2-piecewise regression model [13] was performed to identify whether the correlation between miRNAs and outcomes showed a threshold effect. Once a threshold effect was discovered, we calculated the inflection point, and ORs with 95% CI to the left and right of the inflection point. ORs were adjusted for the same variables as in multivariable logistic analysis. All statistical analyses were conducted using STATA 14.0 (Corporation, College Station, TX, USA), R version 4.1.2 (www.R-project.org, accessed on 1 February 2022) and EmpowerStats (Ver 3.0, http://www.empowerstats.com, X&Y Solutions, Boston, MA, USA, accessed on 1 February 2022). A two-tailed *p* < 0.05 was considered statistically significant.

## 3. Results

### 3.1. Baseline Information

A total of 215 patients (66.7 ± 14.5 years; Female 38.6%) who met the inclusion criteria and consented to participate were included in the analysis. The interval time between stroke onset and blood collection was 25.68 h [IQR 5.75–43.08 h]. All patients completed the three month and one year follow-ups. At 3 months, 23 (10.7%) patients had died, and 106 (49.3%) patients had poor functional outcomes. Among the remaining 192 patients, 6 (3.1%) patients died at 1 year. A total of 88 (40.9%) patients suffered from poor functional outcomes at 1 year. A total of 15 (7%) patients developed spontaneous HT. The median time from stroke onset to spontaneous HT was 91.25 h [IQR 67.9–114.1 h].

### 3.2. Correlation between miRNA Expression Levels and Functional Outcomes

Table 1 showed the baseline characteristics of patients with and without poor functional outcomes at 3 months and 1 year. Patients with poor functional outcomes were older, with lower GCS score, higher NIHSS score, and higher axillary temperature on admission at both 3 months and 1 year (all *p* < 0.05). In addition, most of the patients with poor outcomes at 1 year had cardioembolic stroke, while those without poor outcomes had small artery occlusion stroke. There was no significant difference of miRNA levels between patients with and without 3 month poor outcomes (all *p* > 0.05). The miR-491-5p level was significantly lower in patients with 1-year poor outcomes when compared to those with good outcomes (0.60 [0.27, 1.33] vs. 1.43 [0.31, 7.45], *p* = 0.021). (Figure 1a,b).

We further performed multivariate analysis to explore the independent correlation between miRNA expression level and functional outcomes, adjusted by model 1 (age, GCS on admission, NIHSS on admission, axillary temperature, TOAST classification) and further by model 2 (model 1 + DM, acute myocardial infarction, AIS). In the multivariate logistic analysis, higher miR-491-5p level independently reduced risk of poor functional outcome at 1 year after adjusted by model 1 (OR 0.90 per one-unit Ln-miR-491-5p increase, 95% CI 0.82–0.98, *p* = 0.012) and further by model 2 (OR 0.90 per one-unit Ln-miR-491-5p increase, 95% CI 0.82–0.98, corrected *p* = 0.044). No significant correlation was found between the levels of the other three miRNAs and functional outcome (Table 2).

### 3.3. Correlation between miRNA Expression Levels and Spontaneous HT

Patients who were older and female, had lower admission GCS score, higher admission NIHSS score, and had a history of atrial fibrillation were more likely to develop spontaneous HT (all *p* < 0.05). Patients with spontaneous HT had higher levels of miR-206 on admission when compared to non-HT patients (7.55 [1.35, 50.79] vs. 0.99 [0.28, 5.9], *p* = 0.04) (Appendix A, Figure 1c). There was no significant difference in levels of the other three miRNAs between the HT and non-HT groups.

As shown in Table 3, we further performed multivariate analysis to explore the independent correlation between miRNAs and spontaneous HT, adjusted by model 1 (sex and age), model 2 (model 1 + GCS on admission, NIHSS on admission, AF) and model 3 (model 2 + TOAST classification). After fully adjusted for confounders in model 3, a robust independent correlation was observed between higher miR-206 level and increased risk of spontaneous HT (OR 1.64 per one-unit Ln-miR-206 increase, 95% CI 1.17–2.30, corrected *p* = 0.016).

### 3.4. Analyses of the Nonlinear Correlation between microRNA Levels and Outcomes

The generalized addictive model visualized a nonlinear correlation between miR-491-5p levels and risk of 3-month functional outcome after adjustment for model 2 in Table 2 (Figure 2a). Using the two-piecewise linear regression model, we calculated that the inflection point was 2.10 as the Ln-miR-491-5p level at 3 months. We found that the serum Ln-miR-491-5p level was inversely associated with 3-month poor functional outcome to the left of this point (OR = 0.97 per one-unit Ln-miR-491-5p increase, 95% CI 0.95–0.99, *p* = 0.017), while the correlation was nonsignificant to the right of this point (OR = 0.99 per one-unit Ln-miR-491-5p increase, 95% CI 0.94–1.04, *p* = 0.641) (Table 4).

Similarly, the generalized addictive model showed the non-linear correlation between miR-491-5p level and 1 year poor functional outcome after adjustment for model 2 in Table 2 (Figure 2b). We found that the inflection point of Ln-miR-491-5p value was 2.18. The serum Ln-miR-491-5p level showed a significant reverse correlation with 1 year poor functional outcome to the left of this point (OR = 0.90 per one-unit Ln-miR-491-5p increase, 95% CI 0.86–0.95, *p* < 0.001), while the correlation was nonsignificant to the right of this point (OR = 1.01 per one-unit Ln-miR-491-5p increase, 95% CI 0.94–1.08, *p* = 0.892) (Table 4).

By using generalized addictive model, we identified an approximately linear correlation between higher miR-206 levels and risk of spontaneous HT after adjustment for model 3 in Table 3 (Figure 2c). The inflection point was 2.04. To the left of the inflection point, the OR (95% CI) and *p* value were 1.11 per one-unit Ln-miR-206 increase (1.01–1.03) and 0.009, respectively. However, to the right of the point, the miR-206/spontaneous HT correlation displayed insignificant values: OR = 1.01 per one-unit Ln- miR-206 increase; 95% CI: (0.99–1.04); *p* = 0.384 (Table 4).

## 4. Discussion

In this case control study, we found that higher miR-491-5p expression level could independently reduce the risk of poor functional outcomes at 1 year in AIS patients in a non-linear manner. An increased miR-206 expression level was correlated with a higher risk of spontaneous HT in AIS patients in an approximately linear manner.

MiRNAs have been reported to have the diagnostic potential for stroke [3]. However, few studies have investigated the correlation between serum miRNAs and functional outcome or spontaneous HT. Only five studies performed functional prognostic analysis of the expression of six microRNAs in ischemic stroke patients [14,15,16,17,18]. However, the sample sizes were small, among which only one study obtained a sample over 100 patients [18]. The sampling time from onset, the characteristics and risk factors for stroke patients varied. In addition, evidence of the down-stream regulatory molecules of the previously reported miRNAs is lacking, making it inadequate to explain the underlying mechanism of miRNAs on stroke outcomes. We enrolled a relatively large sample and found that the expression of the MMP-9-related miRNA miR-491-5p could reduce the risk of poor functional outcome after stroke. Few studies have reported the relationship between miRNAs and HT and have mainly focused on patients undergoing reperfusion therapy [19]. A small sample study from our team reported the correlation between miRNAs and spontaneous HT among cardioembolic stroke patients. Here, we showed that the higher level of miR-206 increased the risk of spontaneous HT after AIS.

MMP-9 was reported to relate with functional outcome and spontaneous HT after stroke [5,7,20,21,22]. However, the underlying mechanism is still controversial. MMP-9 inhibition therapy has been reported to have the potential functions of neuroprotection and extension of the thrombolytic window, but its effect remains uncertain. Therefore, the investigation of upstream miRNAs of MMP-9 is necessary. Compared with MMP-9 as a protein, circulating miRNAs are relatively more stable in serum or plasma because they are enclosed within acid- and base-resistant membrane vesicles [18], allowing their quantitation by qRT–PCR. Four miRNAs, miR-21-5p, miR-206, miR-3123 and miR-491-5p, were identified as related to MMP-9 based on three microRNA databases and previous literature [8] and assayed.

To our knowledge, our study is the first to report an independent association of miR-491-5p with functional outcome in stroke patients. MiR-491-5p is a micromolecular noncoding RNA with a length of 20–22 nucleotides [23]. Its accumulation alleviated oxidized low-density lipoprotein-induced (ox-LDL) inflammation, cell viability inhibition, and cell apoptosis [24]. In the development of atherosclerosis (AS), miR-491-5p overexpression alleviates ox-LDL-induced HUVEC injuries by suppressing ICAM1 [24]. MiR491-5p mediates the regulation of MMP-9 expression [25]. It can inhibit the expression of target mRNAs by binding to them, thus exerting a regulatory effect and achieving targeted inhibition of MMP-9 expression [26,27]. The interference of miR-491-5p binding to MMP-9 may confer an increased risk for atherosclerotic cerebral infarction [27]. The existing evidence suggests that miR-491-5p serves as a protective biomarker against endothelial injury in a chronic manner through the regulation of MMP-9 expression. This may help us to understand why a lower level of miR-491-5p in patients results in worse functional outcomes.

MiR-206, a muscle-enriched miRNA (myomiR), has an important role in myogenic differentiation [28]. MiR-206 was reported to be positively correlated with stroke severity [29,30] in humans and infarct volume in an animal model of embolic stroke [31], serving as a detrimental biomarker. Overexpression of miR-206 inhibits neuronal cell viability and proliferation, interfering with neurodevelopment [32]. After stroke onset, miR-206 was reported to disturb angiogenesis by inhibiting vascular endothelial growth factor and hindering the extension of angiogenic signaling to the endothelium [33]. MiR-206 was reported to regulate the secretion of inflammatory cytokines and drive MMP9 expression [34]; MMP-9 was involved in BBB breakdown and HT [5,22,35]. Taken together, miR-206 may serve as a detrimental factor targeting neurons and vasculature in stroke, potentially involved in BBB breakdown through MMP-9 regulation and thus leading to HT. In our study, elevated miR-206 was correlated with a higher risk of spontaneous HT in acute ischemic stroke. Our findings, together with previous studies, suggest miR-206 as a candidate prognostic biomarker for spontaneous HT after stroke.

During the clinical process, one potential restriction of qRT–PCR is that it may require longer than is available for timely recanalization treatment after stroke onset. Patients with acute ischemic stroke within 6 h after symptom onset are highly recommended to assess the indication for reperfusion therapy to improve functional outcome, and the qRT–PCR in this study took over 6 h. Intravenous alteplase therapy or endovascular therapy, if indicated, should not be delayed to await the results of serum miRNA testing. In addition, reperfusion therapy, which increases the risk of HT, may be a distracter and exaggerate the genuine effect of miRNAs on outcomes. For these reasons, we excluded patients receiving reperfusion therapies after stroke onset from our study.

Our study has several strengths. First, we reported the correlation between MMP-9-related miRNAs and functional outcomes, which has rarely been studied before. Second, we focused on the prognostic potential of miRNAs on spontaneous HT, which is common against the background of the underuse of reperfusion therapy, especially in Asians. Third, we constructed a generalized additive model to demonstrate the dose-effect nonlinear correlation between serum miRNA and outcomes, reinforcing our findings of correlation. There are several limitations in our study. First, this was a study based on only one center. In the future, larger, well-designed multicenter studies are needed to verify our results. Second, MMP-9 levels should be sampled simultaneously with miRNAs; the correlation between MMP-9 and miRNAs needs to be further analyzed. We weren’t able to conduct a co-analysis of MMP-9, since the amount of blood sample for each individual patient was inadequate for an additional measurement of MMP-9 after detection for four miRNAs. Nevertheless, our results pave the way for future extensive investigations on the correlation between miRNAs and functional outcomes as well as spontaneous HT after acute ischemic stroke.

## 5. Conclusions

Among AIS patients, a higher miR-491-5p expression level was independently correlated with reduced risk of poor 1 year functional outcomes in a non-linear manner. In addition, an increased miR-206 expression level was correlated with a higher risk of spontaneous HT in an approximately linear manner. Our study showed that miRNAs may become potential biomarkers for poor outcomes in ischemic stroke, helping to guide treatment decisions and the expectations of patients and their families. Further studies on the underlying mechanism are needed to confirm our results.

## Figures and Tables

**Figure 1 brainsci-12-00999-f001:**
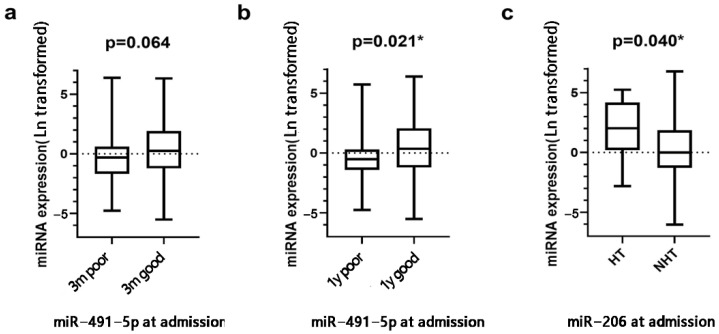
Relative levels of miRNA expression in serum between different outcome groups. (**a**) A non−significant trend of lower miR−491−5p level was observed in patients with 3−month poor outcome. (**b**) Significantly lower miR-491-5p level was observed in patients with 1−year poor outcome. (**c**) Patients with spontaneous HT had higher level of miR−206. The expression levels of miRNAs were performed with the natural log(Ln)-transformed data. The Mann−Whitney U test was used for statistical analysis. 3 m, 3−month; 1 y, 1−year; HT, hemorrhagic transformation; NHT, non−hemorrhagic transformation * *p* < 0.05.

**Figure 2 brainsci-12-00999-f002:**
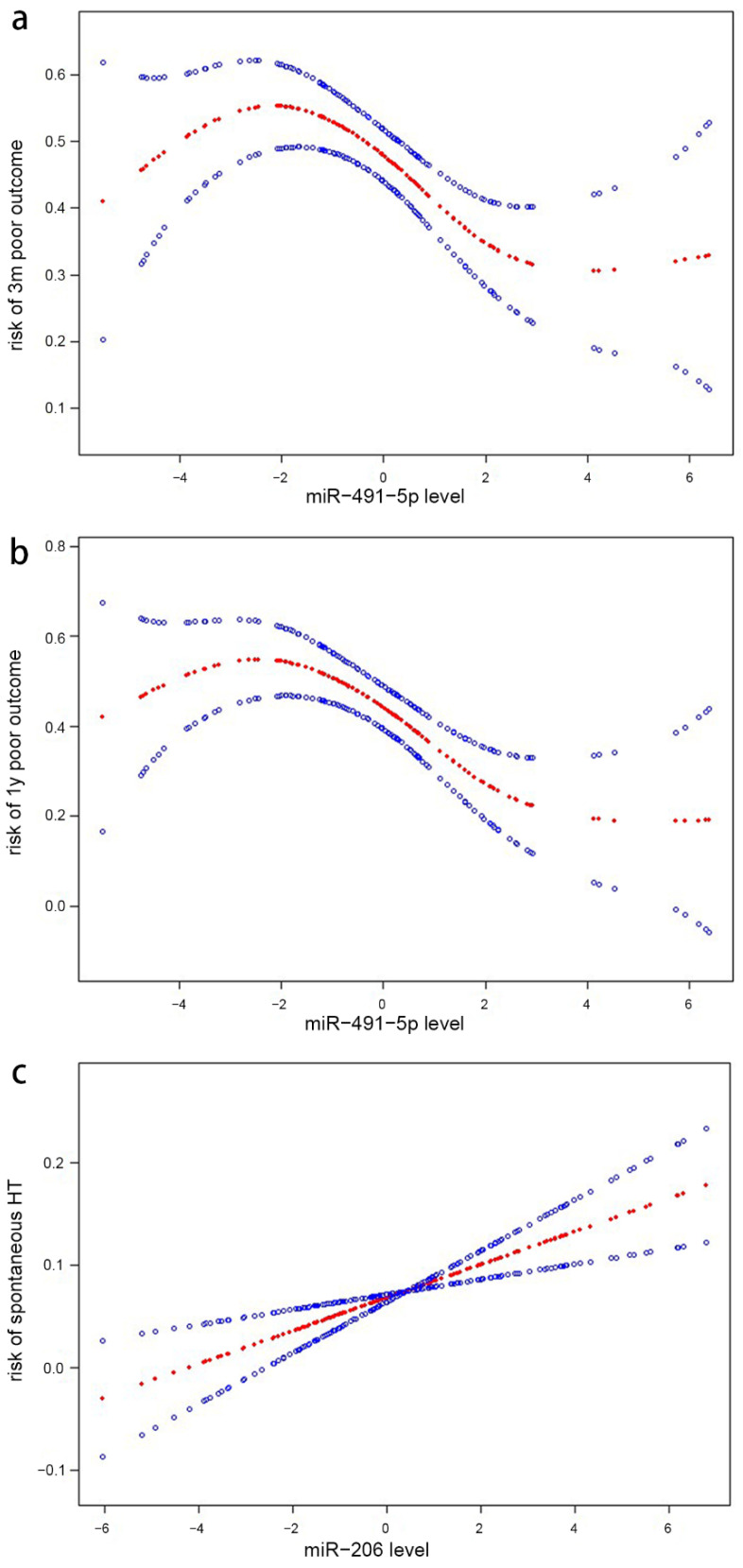
Generalized addictive model (GAM) showed (**a**) the nonlinear correlation between miR−491−5p level and risk of 3-month poor outcome; (**b**) the nonlinear correlation between miR−491−5p level and risk of 1−year poor outcome; (**c**) approximate linear correlation between miR−206 level and risk of spontaneous HT. Red lines stand for odds ratios and blue lines stand for their 95% confidence intervals (CI). The miRNA levels were incorporated into the model as Ln−transformed values of fold difference per µL of serum. The risk of outcomes were calculated and incorporated into the model as Lg(odds ratio). Odds ratios were adjusted for the same variables as model 2 in Table 2 (**a**,**b**) or model 3 in Table 3 (**c**).

**Table 1 brainsci-12-00999-t001:** The baseline characteristics of AIS patients with poor or good outcomes at 3 months and 1 year.

		Overall (*n* = 215)	Poor 3 m Outcome (*n* = 106)	Good 3 m Outcome (*n* = 109)	*p* Value	Poor 1 y Outcome (*n* = 88)	Good 1 y Outcome (*n* = 127)	*p* Value
Age, years, mean (SD)		66.69 (14.52)	69.93 (13.91)	63.53 (14.46)	0.001 *	71.08 (11.53)	61.21 (15.65)	0.001 *
Female, *n* (%)		83 (38.6)	44 (41.5)	39 (35.8)	0.47	50 (56.8)	82 (64.6)	0.315
Onset to admission, h, median [IQR]		21.00 [5.00, 45.00]	21.00 [4.63, 45.00]	21.00 [6.50, 45.00]	0.290	21.00 [2.25, 45.00]	21.00 [8.00, 45.00]	0.280
Onset to blood sampling, h, median [IQR]		25.68 [5.75, 43.08]	21.20 [7.41, 41.27]	27.07 [4.91, 45.22]	0.526	20.94 [4.32, 41.84]	26.64 [7.88, 44.58]	0.228
GCS on admission, median [IQR]		15.00 [13.00, 15.00]	14.00 [10.25, 15.00]	15.00 [14.00, 15.00]	<0.001 *	13.00 [10.00, 15.00]	15.00 [14.00, 15.00]	<0.001 *
NIHSS on admission, median [IQR]		6.00 [3.00, 12.00]	11.00 [4.00, 15.75]	4.00 [2.00, 7.00]	<0.001 *	12.00 [6.00, 16.25]	4.00 [2.00, 7.00]	<0.001 *
Axillary temperature, °C, mean (SD)		36.53 (0.44)	36.64 (0.05)	36.40 (0.03)	<0.001 *	36.67 (0.06)	36.41 (0.03)	<0.001 *
History of risk factors								
Hypertension, *n* (%)		124 (57.7)	60 (56.6)	64 (58.7)	0.861	54 (61.4)	70 (55.1)	0.441
DM, *n* (%)		53 (24.7)	29 (27.4)	24 (22.0)	0.453	28 (31.8)	25 (19.7)	0.062
Hyperlipidemia, *n* (%)		8 (3.7)	3 (2.8)	5 (4.6)	0.749	4 (4.5)	4 (3.1)	0.869
AF, *n* (%)		31 (14.4)	18 (17.0)	13 (11.9)	0.389	17 (19.3)	14 (11.0)	0.132
Acute myocardial infarction, *n* (%)		4 (1.9)	4 (3.8)	0 (0.0)	0.123	4 (4.5)	0 (0.0)	0.056
Valvular heart disease, *n* (%)		11 (5.1)	6 (5.7)	5 (4.6)	0.962	6 (6.8)	5 (3.9)	0.53
Transient ischemic attack, *n* (%)		6 (2.8)	3 (2.8)	3 (2.8)	1	2 (2.3)	4 (3.1)	1
AIS, *n* (%)		43 (20.0)	23 (21.7)	20 (18.3)	0.658	23 (26.1)	20 (15.7)	0.089
Hemorrhagic stroke, *n* (%)		3 (1.4)	1 (0.9)	2 (1.8)	1	2 (2.3)	1 (0.8)	0.748
Therapies before admission								
Antiplatelet therapy, *n* (%)		28 (13.0)	14 (13.2)	14 (12.8)	1	13 (14.8)	15 (11.8)	0.668
Lipid lowering, *n* (%)		19 (8.8)	8 (7.5)	11 (10.1)	0.677	7 (8.0)	12 (9.4)	0.892
Anticoagulant therapy, *n* (%)		10 (4.7)	3 (2.8)	7 (6.4)	0.354	4 (4.5)	6 (4.7)	1
TOAST classification, *n* (%)					0.089			0.005 *
LAA		62 (28.8)	33 (31.1)	29 (26.6)		28 (31.8)	34 (26.8)	
SAO		49 (22.8)	16 (15.1)	33 (30.3)		10 (11.4)	39 (30.7)	
CE		58 (27.0)	34 (32.1)	24 (22.0)		32 (36.4)	26 (20.5)	
SOE		5 (2.3)	3 (2.8)	2 (1.8)		1 (1.1)	4 (3.1)	
SUE		41 (19.1)	20 (18.9)	21 (19.3)		17 (19.3)	24 (18.9)	
ECASS classification, *n* (%)					0.27			0.762
No		200 (93.0)	95 (89.6)	105 (96.3)		80 (90.9)	120 (94.5)	
HI1		6 (2.8)	4 (3.8)	2 (1.8)		3 (3.4)	3 (2.4)	
HI2		5 (2.3)	4 (3.8)	1 (0.9)		3 (3.4)	2 (1.6)	
PH1		4 (1.9)	3 (2.8)	1 (0.9)		2 (2.3)	2 (1.6)	
PH2		0 (0.0)	0 (0.0)	0 (0.0)		0 (0.0)	0 (0.0)	
Levels of miRNA expression, fold difference								
miR-21-5p, median [IQR]		0.82 [0.18, 4.10]	0.89 [0.17, 4.66]	0.77 [0.21, 2.53]	0.635	0.69 [0.17, 2.23]	0.88 [0.17, 6.10]	0.363
miR-491-5p, median [IQR])		0.98 [0.29, 2.45]	0.72 [0.21, 1.62]	1.25 [0.30, 4.98]	0.064	0.60 [0.27, 1.33]	1.43 [0.31, 7.45]	0.021 *
miR-3123, median [IQR]		1.60 [0.08, 12.26]	1.11 [0.04, 15.79]	1.92 [0.15, 10.71]	0.755	0.78 [0.02, 9.23]	1.98 [0.09, 14.37]	0.326
miR-206, median [IQR]		1.15 [0.29, 7.57]	1.51 [0.35, 7.47]	0.84 [0.25, 9.00]	0.476	1.75 [0.36, 9.85]	1.95 [0.51, 26.62]	0.414

AIS, acute ischemic stroke; 3 m, 3-month; 1 y, 1-year; SD, standard deviation; IQR, interquartile range; GCS, Glasgow Coma Scale; NIHSS, National Institutes of Health Stroke Scale; DM, diabetes mellitus; AF, atrial fibrillation; TOAST, Trial of Org 10,172 in Acute Stroke Treatment; LAA, large-artery atherosclerosis; SAO, small-artery occlusion; CE, cardioembolism; SOE, acute stroke of other determined etiology; SUE, stroke of underdetermined etiology; ECASS, The European Cooperative Acute Stroke Study; HI, hemorrhagic information; PH, parenchymal hemorrhagic. * *p* < 0.05.

**Table 2 brainsci-12-00999-t002:** Unadjusted and adjusted odds ratios of miRNAs expression for poor outcomes at 3 months and 1 year.

	3 Months						
	Unadjusted		Model 1		Model 2		
	OR (95% CI)	*p* Value	OR (95% CI)	*p* Value	OR (95% CI)	*p* Value	Corrected *p* Value ^#^
miR-21-5p ^†^	1.01 (0.92–1.12)	0.785	1.00 (0.95–1.06)	0.947	1.00 (0.95–1.05)	0.992	0.992
miR-491-5p ^†^	0.90 (0.77–1.05)	0.177	0.93 (0.85–1.02)	0.145	0.93 (0.85–1.03)	0.147	0.588
miR-3123 ^†^	0.97 (0.88–1.07)	0.545	1.00 (0.95–1.04)	0.901	1.00 (0.95–1.05)	0.958	0.992
miR-206 ^†^	1.03 (0.91–1.16)	0.684	1.01 (0.96–1.07)	0.640	1.01 (0.95–1.07)	0.783	0.992
	**1 Year**						
	**Unadjusted**		**Model 1**		**Model 2**		
	**OR (95% CI)**	***p* Value**	**OR (95% CI)**	***p* Value**	**OR (95% CI)**	***p* Value**	**Corrected *p* Value ^#^**
miR-21-5p ^†^	0.99 (0.900–1.100)	0.921	0.98 (0.93–1.03)	0.499	0.98 (0.93–1.03)	0.456	0.347
**miR-491-5p ^†^**	0.84 (0.71–0.99)	**0.036 ***	0.90 (0.82–0.98)	**0.012 ***	0.90 (0.82–0.98)	**0.014 ***	**0.044 ***
miR-3123 ^†^	0.93 (0.84–1.03)	0.167	0.97 (0.92–1.02)	0.282	0.98 (0.93–1.03)	0.348	0.347
miR-206 ^†^	0.98 (0.87–1.10)	0.712	0.98 (0.92–1.05)	0.570	0.98 (0.84–1.15)	0.476	0.347

OR, odds ratio; CI: confidence interval; Model 1 was adjusted by age, GCS on admission, NIHSS on admission, axillary temperature, TOAST classification; Model 2 was adjusted by model 1 + DM, acute myocardial infarction, AIS; ^†^ Natural log(Ln)-transformed values of fold difference per µL of serum were used; ^#^ *p* values were corrected by Benjamini-Hochberg method; * *p* value < 0.05.

**Table 3 brainsci-12-00999-t003:** Unadjusted and adjusted odds ratios of miRNAs expression for HT.

	Unadjusted		Model 1		
	OR (95% CI)	*p* Value	OR (95% CI)	*p* Value	
miR-21-5p ^†^	0.98 (0.81–1.19)	0.848	0.93 (0.76–1.13)	0.463	
miR-491-5p ^†^	1.05 (0.80–1.38)	0.725	1.01 (0.76–1.34)	0.932	
miR-3123 ^†^	0.91 (0.75–1.09)	0.289	0.91 (0.76–1.09)	0.324	
**miR-206** ^†^	1.25 (1.00–1.57)	**0.048 ***	1.45 (1.09–1.94)	**0.012 ***	
	**Model 2**		**Model 3**		
	**OR (95% CI)**	***p* Value**	**OR (95% CI)**	***p* Value**	**Corrected *p* Value ^#^**
miR-21-5p ^†^	0.88 (0.71–1.09)	0.259	0.85 (0.67–1.07)	0.165	0.330
miR-491-5p ^†^	1.04 (0.77–1.42)	0.784	1.08 (0.78–1.49)	0.649	0.649
miR-3123 ^†^	0.91 (0.76–1.10)	0.338	0.93 (0.77–1.13)	0.466	0.621
**miR-206** ^†^	1.61 (1.16–2.23)	**0.004 ***	1.64 (1.17–2.30)	**0.004 ***	**0.016 ***

HT, hemorrhagic transformation; OR, odds ratio; CI: confidence interval; Model 1 was adjusted by sex and age; Model 2 was adjusted by model 1 + GCS on admission, NIHSS on admission, AF; Model 3 was adjusted by model 2 + TOAST classification; ^†^ Natural log(Ln)-transformed values of fold difference per µL of serum were used; ^#^ *p* values were corrected by Benjamini-Hochberg method; * *p* value < 0.05.

**Table 4 brainsci-12-00999-t004:** The results of the two piecewise linear regression model.

Inflection Points of miRNA Level ^†^	OR (95% CI)	*p* Value
**miR-491-5p ^#^**	**3-month poor outcome**	
<2.10	0.97 (0.95–0.99)	0.017 *
>2.10	0.99 (0.94, 1.04)	0.641
**miR-491-5p ^#^**	**1-year poor outcome**	
<2.18	0.90 (0.86, 0.95)	<0.001 *
>2.18	1.01 (0.94, 1.08)	0.892
**miR-206 ^§^**	**Spontaneous HT**	
<2.04	1.02 (1.01, 1.03)	0.009 *
>2.04	1.01 (0.99, 1.04)	0.384

HT, hemorrhagic transformation; OR, odds ratio; CI: confidence interval; ^†^ Natural log(Ln)-transformed values of fold difference per µL of serum were used; ^#^: Adjustment: age, GCS on admission, NIHSS on admission, axillary temperature, TOAST classification, DM, acute myocardial infarction, AIS; **^§^**: Adjustment: sex, age, GCS on admission, NIHSS on admission, AF, TOAST classification; * *p* < 0.05.

## Data Availability

The data that support the findings of this study are available from the corresponding author upon reasonable request.

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
