# Peer review of "Serum microRNA miR-491-5p/miR-206 Is Correlated with Poor Outcomes/Spontaneous Hemorrhagic Transformation after Ischemic Stroke: A Case Control Study"

_brainsci, 2022, doi:10.3390/brainsci12080999_

Round 1

Reviewer 1 Report

The authors performed an analysis of the correlation of miR-491-5p/miR-206 on the one hand with good prognosis and on the other hand with the probability of developing hemorrhagic transformation.

It is a well-executed work with concise results. This reviewer has only a series of formatting and methodological recommendations.

In the abstract include the number of patients in the methods section.

It would be convenient for the reviewer to know if it is a retrospective study or in the case of a prospective study it would be possible to add a series of co-analysis of MMP or metabolites of interest to confirm the hypothesis supported by an increase or decrease of circulating MiRNA.

This reviewer recommends increasing the size of Figure 1 to allow a clear evaluation of the results.

Regarding the variables considered, it would be important to include the axillary temperature of the patients; since hyperthermia is a process associated with a worse prognosis of the patients and is also related to the functionality of MMPs.

The discussion is well presented and analyzes the results correctly.

Author Response

Point 1: In the abstract include the number of patients in the methods section.

Response 1: Thank you for pointing this out. We have added the number of patients in the methods section of abstract(Line 16).

Point 2: It would be convenient for the reviewer to know if it is a retrospective study or in the case of a prospective study it would be possible to add a series of co-analysis of MMP or metabolites of interest to confirm the hypothesis supported by an increase or decrease of circulating MiRNA.

Response 2:

Thank you for your constructive suggestion. In this study,we retrospectively measured the miRNA levels of AIS patients, whose blood samples and clinical information were prospectivelly collected during hospitalization and follow-ups. We have detailed this information in Methods section(Line 60-61, Line 68-82, Line 85-88, Line 95).
We did plan to conduct a co-analysis of MMP-9, however, the amount of blood sample for each individual patient was inadquate for an additional measurement of MMP-9 after detection for four miRNAs. We have discussed this limitation in Discussion section(Line 386-390). On the other hand, previous literatures have reported the involvement of miR-491-5p and miR-206 in regulation of MMP-9 expression (DOI: 10.3892/ijmm.2021.5050, DOI: 10.1016/j.bbrc.2016.06.038), as well as the correlation between MMP-9 and functional outcomes/HT (DOI:10.3389/fneur.2020.523506, DOI: 10.1161/STROKEAHA.106.481556), which could support our  hypothesis to some extent. In future studies, we will conduct co-analysis of MMPs and other related  metabolites in animal experiments and multicenter cohort studies with larger samples to confirm our result.

Point 3: This reviewer recommends increasing the size of Figure 1 to allow a clear evaluation of the results.

Response 3: We are sorry for out negligence. We have increased the size of Figure 1 to allow a clear evaluation of the results(Line 194).

Point 4: Regarding the variables considered, it would be important to include the axillary temperature of the patients; since hyperthermia is a process associated with a worse prognosis of the patients and is also related to the functionality of MMPs.

Response 4: Thank you for your constructive suggestion. We have added the axillary temperature of the patients for univariate analysis, and included this factor as confounders for adjustment in multivariate analysis as well. (Line 71, Line 167, Table 1, Line 209-223, Table 2, Table S1)

Reviewer 2 Report

In the manuscript the authors investigated relationships between serum microRNA levels (miR-491-5p; miR-206; miR-21-5p and miR-3123) and functional outcome in acute ischemic stroke measured using modified Rankin Scale as well as with probability of spontaneous hemorrhagic transformation. This is an interesting study which extends the previous work of the same group, published in 2019 (https://doi.org/10.3389/fneur.2019.00945). Compared to the previous work the number of patients is significantly increased and the sample includes patients of different ethiologies and not only cardioembolic stroke. However, several questions should be clarified and detailed before publication.

- Lines 127 and 308-310: the authors claim that they have constructed a generalized additive model but neither any details on how it was created nor the resulting model is described. Since it is one of the cornerstones for the present work, this aspect should be described in detail.

- Fig.2 How the rist of spontaneous HT was calculated?

- Fig.1 Please, add “at 3 month”, “at 1 year”, “at admission” to the figure itself. Otherwise, the figure looks confusing unless you read the legend very carefully.

- I wonder whether the authors have performed any corrections for multiple comparisons (the number of comparisons between the groups is quite large) and whether the detected differences for microRNA would survive such corrections?

- Lines 171-179: Please describe how the mentioned multivariate analysis was performed.

Author Response

Point 1: Lines 127 and 308-310: the authors claim that they have constructed a generalized additive model but neither any details on how it was created nor the resulting model is described. Since it is one of the cornerstones for the present work, this aspect should be described in detail.

Response 1: Thank you for pointing this out. The generalized additive model was used to visually confirm the relationship between the expression levels of miRNAs and risk of outcomes. The miRNA expression levels were incorporated into the model as Log 10(Lg)-transformed values of normalized copy numbers per µL of serum. The risk of outcomes were calculated and incorporated into the model as Lg(odds ratio). Odds ratios were adjusted for the same variables as in multivariable logistic analysis. We have added the details in Methods section, Results section, and figure legends(Fig2). (Line 140-144, Line 266-272, Line 301-302, Line 309-311, Fig 2)

Point 2: Fig.2 How the risk of spontaneous HT was calculated?

Response 2: Thank you for pointing this out. The risk of spontaneous HT was calculated as Lg(odds ratio). Odds ratios were adjusted for the same variables as in multivariable logistic analysis. We have detailed the calculation of risk in Methods section, Results section, and figure legends(Fig2). (Line 140-144, Line 309-311, Fig 2)

Point 3: Fig.1 Please, add “at 3 month”, “at 1 year”, “at admission” to the figure itself. Otherwise, the figure looks confusing unless you read the legend very carefully.

Response 3: We are sorry for out negligence. We have added “3-month”, “1-year”and “at admission” at appropriate locations in Fig.1(Line 194)

Point 4: I wonder whether the authors have performed any corrections for multiple comparisons (the number of comparisons between the groups is quite large) and whether the detected differences for microRNA would survive such corrections?

Response 4: Thank you for your constructive suggestion. We performed correction for multiple comparisons using Benjamini-Hochberg method(mentioned in Methods section, Line 138-139) and displayed the corrected P values in the Table 2 and Table 3(Line 225, Line 256). The detected differences for miR-491-5p and miR-206 survived after corrections.

Point 5Lines 171-179: Please describe how the mentioned multivariate analysis was performed.

Response 5: Thank you for pointing this out. We have added the description of how the multivariate analysis was performed in Methods section, Results section and table legends(Line 134-138, Line 209-223, Table 2, Line 246-255, Table 3).

Round 2

Reviewer 2 Report

All comments were resolved.